# PIEClass: Weakly-Supervised Text Classification with Prompting and Noise-Robust Iterative Ensemble Training

**Yunyi Zhang, Minhao Jiang, Yu Meng, Yu Zhang, Jiawei Han**
University of Illinois Urbana-Champaign, IL, USA
{yzhan238, minhaoj2, yumeng5, yuz9, hanj}@illinois.edu

## Abstract

Weakly-supervised text classification trains a classifier using the label name of each target class as the only supervision, which largely reduces human annotation efforts. Most existing methods first use the label names as static keyword-based features to generate pseudo labels, which are then used for final classifier training. While reasonable, such a commonly adopted framework suffers from two limitations: (1) keywords can have different meanings in different contexts and some texts may not explicitly contain any keyword, so keyword matching can induce noisy and inadequate pseudo labels; (2) the errors made in the pseudo label generation stage will directly propagate to the classifier training stage without a chance of being corrected. In this paper, we propose a new method, PIEClass, consisting of two modules: (1) a pseudo label acquisition module that uses zero-shot prompting of pre-trained language models (PLM) to get pseudo labels based on contextualized text understanding beyond static keyword matching, and (2) a noise-robust iterative ensemble training module that iteratively trains classifiers and updates pseudo labels by utilizing two PLM fine-tuning methods that regularize each other. Extensive experiments show that PIEClass achieves overall better performance than existing strong baselines on seven benchmark datasets and even achieves similar performance to fully-supervised classifiers on sentiment classification tasks.[1]

## 1 Introduction

Text classification is a fundamental NLP task with a wide range of downstream applications, such as question answering (Rajpurkar et al., 2016), sentiment analysis (Tang et al., 2015), and event detection (Zhang et al., 2022c). Earlier studies train text classifiers in a fully-supervised manner that requires a substantial amount of training data (Zhang et al., 2015; Yang et al., 2016), which are generally costly to obtain. To eliminate the need for labeled training samples, weakly-supervised text classification settings (Meng et al., 2018, 2020; Wang et al., 2021) are proposed, which aim to train text classifiers using the label names of target classes as the only supervision. Such settings are intriguing especially when obtaining high-quality labels is prohibitively expensive.

Recent advancements in large generative language models (LLMs) (e.g., ChatGPT, GPT-4 (OpenAI, 2023)) make it a valid approach to directly prompt them in a zero-shot manner for text classification without labeled data. For example, people may provide a restaurant review and ask an LLM "What is the sentiment of this document?", and the model will generate an answer according to its understanding. However, there are certain limitations of this method for the weakly-supervised text classification setting. First, directly prompting LLMs cannot utilize any domain-specific information hidden in the unlabeled data, because it is intractable to fine-tune such a large model and the prompts can hardly incorporate any corpus-level information, especially for corpora not appearing in LLMs' pre-training data (e.g., private domains). Second, deploying LLMs is expensive, while many text classification applications require fast real-time inference (e.g., email and review classification).

Another line of studies tailored for weakly-supervised text classification aims to train a moderate-size classifier with a task-specific *unlabeled* corpus. Given the label names, these methods first acquire class-indicative keywords using PLMs (Meng et al., 2020; Wang et al., 2021) or corpus-level co-occurrence features (Zhang et al., 2021, 2022b). The keywords are then used as static features to generate pseudo-labeled documents for fine-tuning the final classifier. De-

---

[1] Code can be found at https://github.com/yzhan238/PIEClass.

spite their promising performance, the aforementioned weakly-supervised methods may suffer from two major limitations. First, these methods are keyword-driven by using class-indicative keywords as static context-free features to generate pseudo labels with different forms of string matching. However, some texts may not contain such class-indicative keywords and keywords may have different meanings in different contexts, so using them as static features can lead to noisy and inadequate pseudo labels. Such an issue is more serious for abstract classes like sentiments that require understanding rhetoric. For example, a food review *"It is to die for!"* contains the keyword *"die"* which itself is negative, but the entire review expresses a strong positive sentiment, and keyword-driven methods will likely struggle in these cases. Second, most existing methods are two-stage by conducting pseudo label acquisition and text classifier training in two successive steps. Although different pseudo label acquisition methods are explored to improve their quality (e.g., masked language modeling (Meng et al., 2020), clustering of PLM embeddings (Wang et al., 2021), or large textual entailment models trained with external data (Park and Lee, 2022)), there is still a large performance gap between weakly-supervised and fully-supervised settings, because erroneous pseudo labels in the first stage will propagate to and harm the classifier training stage without a chance to be corrected.

To address the limitations of existing works, in this paper, we propose PIEClass: **P**rompting and **I**terative **E**nsemble Training for Weakly-Supervised Text **Class**ification. PIEClass consists of two modules. (1) Pseudo label acquisition via PLM prompting. By designing a task-specific prompt, we can apply a moderate-size PLM to infer the class label of documents based on the entire input sequence, which is thus contextualized and beyond static keyword features. In this work, besides the well-studied prompting method using PLMs pre-trained with the masked language modeling task (MLM) (e.g., BERT (Devlin et al., 2019), RoBERTa (Liu et al., 2019)), we also explore a different prompting method for a discriminative pre-trained language model, ELECTRA (Clark et al., 2020), and compare them in the experiments. (2) Noise-robust training with iterative ensemble. In each iteration, we train text classifiers using the current pseudo labels and then use the confident predictions to re-select the pseudo labels. In this

way, we can gradually expand the pseudo labels which can be used to train better text classifiers. To avoid erroneous pseudo labels accumulated during the iterative process, we propose to utilize two PLM fine-tuning strategies, head token fine-tuning and prompt-based fine-tuning, as two complementary views of the data: One captures the semantics of the entire sequence while the other interprets the contexts based on the prompts. We use the two views to regularize each other and further apply model ensemble to improve the noise robustness of the pseudo label expansion process.

To summarize, the contributions of this paper are as follows: (1) We propose to use the contextualization power of PLM prompting to get pseudo labels for the weakly-supervised text classification task instead of static keyword-based features. (2) We explore the prompting method of a discriminative PLM on the text classification task and compare it with prompting methods for MLM. (3) We propose a noise-robust iterative ensemble training method. To deal with noisy pseudo labels, we utilize two PLM fine-tuning strategies that regularize each other and apply model ensemble to enhance the pseudo label quality. (4) On seven benchmark datasets, PIEClass overall performs better than strong baselines and even achieves similar performance to fully-supervised methods.

## 2 Problem Definition

The weakly-supervised text classification task aims to train a text classifier using label names as the only supervision. Formally, given a set of documents $\mathcal{D} = \{d_1, \ldots, d_n\}$ and $m$ target classes $\mathcal{C} = \{c_1, \ldots, c_m\}$ with their associated label names $l(c)$, our goal is to train a text classifier $F$ that can classify a document into one of the classes. For example, we may classify a collection of news articles using the label names "politics", "sports", and "technology". Notice that, there are previous studies using more than one topic-indicative keyword or a few labeled documents as supervision, whereas here, we follow the *extremely weak supervision* setting (Wang et al., 2021) and only use the sole surface name of each class as supervision.

## 3 Methodology

To address the limitations of existing methods for weakly-supervised text classification, we introduce our method, PIEClass, in this section, which contains two major modules: (1) *zero-shot prompting*

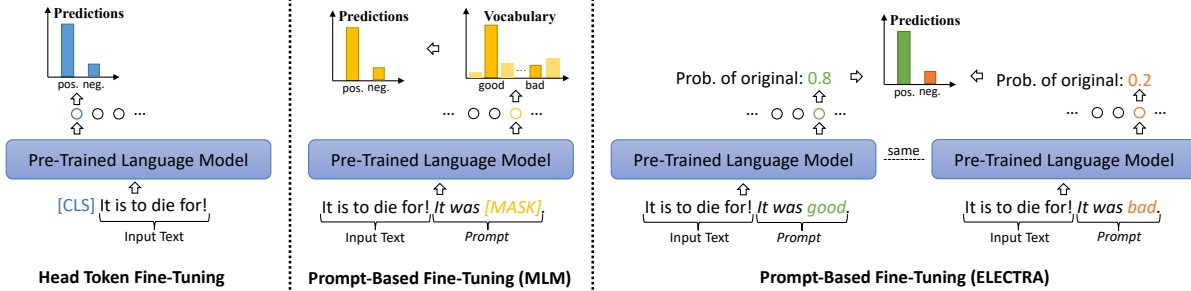

Figure 1: Examples of different fine-tuning strategies on the text classification task. (left) Head token fine-tuning randomly initializes a linear classification head and directly predicts class distribution using the [CLS] token, which needs a substantial amount of training data. (middle) Prompt-based fine-tuning for MLM-based PLM converts the document into the masked token prediction problem by reusing the pre-trained MLM head. (right) Prompt-based fine-tuning for ELECTRA-style PLM converts documents into the replaced token detection problem by reusing the pre-trained discriminative head. Given a document, one input sequence is constructed for each label name.

*for pseudo label acquisition* and (2) *noise-robust training with iterative ensemble*. Figure 2 shows an overview of PIEClass.

## 3.1 Zero-Shot Prompting for Pseudo Label Acquisition

Most existing weakly-supervised text classification methods use a set of static class-indicative keywords to assign pseudo labels to unlabeled documents based on either direct string matching (Meng et al., 2018) or static class embeddings (Wang et al., 2021). However, keywords can only provide limited supervision with low coverage, given that most of the documents do not contain any class-indicative keywords. Also, a document containing keywords does not necessarily indicate that it belongs to the corresponding class because the keywords can mean differently in different contexts. Such issues are more serious for abstract classes that involve more rhetoric, such as sentiment classification. For example, a food review *"It is to die for!"* does not have any single keyword indicating the positive sentiment and even contains the word *"die"* that seems negative, but we can still infer its strong positive sentiment based on our contextualized text understanding beyond static keywords.

To tackle the problem of existing methods and acquire pseudo labels beyond context-free keyword features, we propose to apply *zero-shot prompting* of PLMs. The prompt-based method aims to close the gap between the pre-training task of PLM and its downstream applications, so we can directly use a pre-trained but not fine-tuned PLM with prompts to get high-quality pseudo labels for the text classification task. Also, prompting the PLM guides it to understand the entire context, and thus its predictions are contextualized. Figure 1 (left and middle)

shows examples of standard head token fine-tuning and the popular prompting method for MLM.

Besides utilizing the MLM-based prompting method, in this work, we propose to exploit a discriminative PLM, ELECTRA (Clark et al., 2020), for zero-shot prompting. During pre-training, ELECTRA uses an auxiliary model to generate training signals and trains the main model to denoise it. More specifically, a small Transformer model called a "generator" is trained with masked language modeling to replace the randomly masked tokens of the input text, and then the main Transformer model called a "discriminator" is trained to predict whether each token in the corrupted example is original or replaced by the generator (Clark et al., 2020).

Recent studies have shown the potential of ELECTRA in prompt-based methods (Xia et al., 2022; Yao et al., 2022; Li et al., 2022). Figure 1 (right) shows an example. To generalize the usage of ELECTRA-based prompting to weakly-supervised text classification, we can fill in a template $\mathcal{T}^{\text{ELECTRA}}$ with a document $d$ and one of the label names $l(c)$. The template is designed in a way that the correct label name should be the "original" token of this constructed input while the wrong ones are "replaced". Take the sentiment classification task as an example. If we want to classify whether a restaurant review $d$ expresses a positive or negative sentiment given their label names "good" and "bad", we can construct the following two input sequences to ELECTRA,

$$\mathcal{T}^{\text{ELECTRA}}(d, \text{good}) = d \text{ It was \underline{good}.}$$
$$\mathcal{T}^{\text{ELECTRA}}(d, \text{bad}) \ = d \text{ It was \underline{bad}.}$$

The constructed inputs are individually fed into a pre-trained ELECTRA discriminator and its dis-

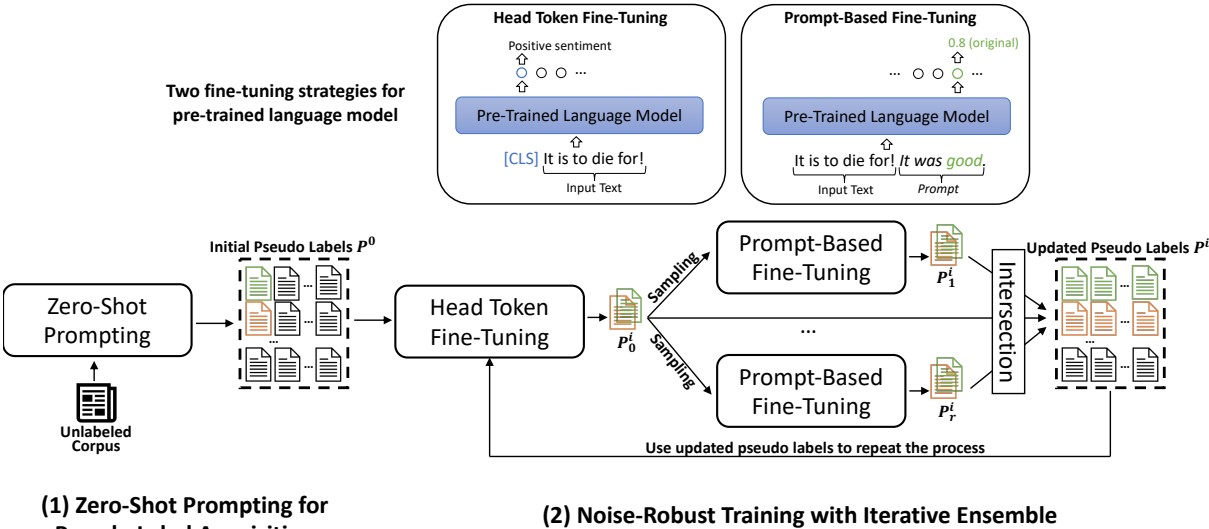

Figure 2: Overview of the PIEClass framework.

criminative classification head $f$ to get the probability of being original for each label name,

$$p(l(c)|d) = \text{Sigmoid}(f(\mathbf{h}^{l(c)})), \quad (1)$$

where $\mathbf{h}^{l(c)}$ is the contextualized representation of the label name $l(c)$ in this context. The confidence of document $d$ belonging to a class $c$ is the normalized probability,

$$p(c|d) = \frac{p(l(c)|d)}{\sum_{c' \in \mathcal{C}} p(l(c')|d)}. \quad (2)$$

After getting the predictions for all the documents in $\mathcal{D}$, we take the top $t^0$ percentage of the documents with the highest confidence scores as our initial pseudo label pool $\mathcal{P}^0$.

### 3.2 Noise-Robust Training with Iterative Ensemble

With the initial pseudo labels, existing methods directly fine-tune (using the head token) a text classifier with such labels to get the final classifier. However, since the pseudo labels are noisy with typical noise ratios ranging from 15% to 50% (Mekala et al., 2022), the performance of the final classifier is limited by the quality of pseudo labels, leading to a large performance gap between weakly-supervised and fully-supervised settings. Therefore, inspired by the self-training method for semi-supervised learning, we propose to iteratively train a text classifier and use its confident predictions to find more high-quality pseudo labels, which can help to train an even better classifier.

However, unlike semi-supervised settings where the initial labels are perfect, here we only have noisy pseudo labels $\mathcal{P}^0$ from the last step. When we train a text classifier with noisy data as supervision, the classifier will likely predict those

wrongly labeled samples wrong with high confidence again. Therefore, if we strictly follow the standard self-training method, the noise will stay and accumulate in the pseudo label pool. To tackle such a challenge, we propose an *iterative ensemble training* method with two types of ensemble to ensure the quality of pseudo labels. First, we utilize two PLM fine-tuning methods, head token and prompt-based fine-tuning, to train classifiers individually in each iteration. Here, the head token fine-tuning behaves like a *sequence-level view* of documents by capturing the information of the entire input document, while prompt-based fine-tuning serves as a *token-level view* by focusing more on the context surrounding the label name (or masked token if using MLM) in the prompt. The two views can complement each other to better exploit the power of PLMs. Second, since the prompt-based method converts the downstream task into the same form as the pre-training task and reuses the pre-trained classification head, it only requires a small amount of data to achieve competitive performance with head token fine-tuning. This allows us to further apply model ensemble by fine-tuning multiple individual prompt-based classifiers to further improve the noise-robust (Laine and Aila, 2017; Meng et al., 2021). Finally, in each iteration, we freshly initialize the classifiers with pre-trained weights, re-select all the pseudo labels, and only keep the top predictions agreed upon by all the classifiers to ensure the quality. Our idea shares a similar spirit as co-training (Blum and Mitchell, 1998). The major difference is that standard co-training learns from initial *clean data* and utilizes two data views to progressively label unlabeled

data, whereas our method does not have access to annotated training data but instead uses two data views as regularization along with model ensemble to improve model's noise-robustness trained on *pseudo-labeled data*.

Specifically, for iteration $i$, we first use the head token to fine-tune a text classifier, $F_0^i : \mathcal{D} \rightarrow \mathcal{C}$, using the current pseudo labels $\mathcal{P}^{i-1}$ in a fully-supervised way. After training, we use the classifier to make a prediction on each document to get $(d_j, F_0^i(d_j))$ and a confidence score $cf_0^i(d_j)$ which is the normalized probability of prediction $F_0^i(d_j)$. Then, we will rank the predictions based on their confidence scores and select the top $t_i$ percentage of them whose confidence scores are greater than a threshold $p$ as candidate pseudo labels $\mathcal{P}_0^i$. The threshold $t_i$ is linearly increasing with iteration numbers, $t_i = i \cdot s$, where $s$ is a hyperparameter. We use an increasing threshold because if we keep the threshold constant, the pseudo samples in the last iteration will be predicted confidently again, making the pseudo samples almost the same in the iterative process and the classifiers overfit more to the limited number of pseudo samples [2].

Because $\mathcal{P}_0^i$ can be noisy, we then utilize prompt-based fine-tuning as a second view to improve the quality of pseudo labels. We randomly sample $r$ subsets of $\mathcal{P}_0^i$, $S_k$, each of size $q \cdot |\mathcal{P}_0^i|$, $q \in (0, 1)$ and fine-tune $r$ classifiers $F_k^i$, $k \in \{1, \ldots, r\}$, using prompt-based fine-tuning. With a small sampling ratio $q$, the noisy labels are unlikely to be sampled repeatedly into different subsets, so this sampling process will further improve the noise robustness of model ensemble. To fine-tune ELECTRA-style PLMs using prompts, each data sample $d$ will have $|\mathcal{C}|$ individual input sequences $\{\mathcal{T}^{\text{ELECTRA}}(d, l(c))\}_{c \in \mathcal{C}}$, and the target class $F_0^i(d)$ should be predicted as "original" while all the others "replaced". The model is trained with binary cross entropy loss

$$\mathcal{L}^{\text{ELECTRA}} = - \sum_{d \in S_k} \Big( \log p(F_0^i(d)|d) + \sum_{c' \neq F_0^i(d)} \log \big( 1 - p(c'|d) \big) \Big). \quad (3)$$

After training, we follow the same process as the classifier $F_0^i$ to select the top $t_i$ percentage of most confident predictions by each classifier $F_k^i$ as candidate pseudo labels $\mathcal{P}_k^i$. Finally, we take the intersection of all the candidate pseudo labels as the final pseudo label pool for this iteration,

---

**Algorithm 1:** PIEClass

**Input:** A corpus $\mathcal{D}$; a set of classes $\mathcal{C}$ and their label names $l(c)$, $c \in \mathcal{C}$; a pre-trained language model $E$; a template $\mathcal{T}$ for prompting.
**Output:** A text classifier $F$ for classes $\mathcal{C}$.

1   // Zero-Shot Prompting for Pseudo Label Acquisition;
2   **for** $d \in \mathcal{D}$ **do**
3     **for** $c \in \mathcal{C}$ **do**
4       $\mathcal{T}(d, l(c)) \leftarrow$ Construct input with the template;
5       $p(l(c)|d) \leftarrow$ Prompt $E$ with Eq. (1);
6     $p(c|d) \leftarrow$ Eq. (2);
7   $P^0 \leftarrow$ top $t^0$ percentage of predictions;
8   // Noise-Robust Training with Iterative Ensemble;
9   **for** $i \leftarrow 1$ *to* $T$ **do**
10    $F_0^i \leftarrow$ Head token fine-tuning using $P^{i-1}$;
11    $P_0^i \leftarrow$ Select top $t_i$ predictions by $F_0^i$;
12    $\mathcal{S} \leftarrow$ Randomly sample $r$ subsets of $P_0^i$;
13    **for** $S_k \in \mathcal{S}$ **do**
14     $F_k^i \leftarrow$ Prompt-based fine-tuning using $S_k$;
15     $P_k^i \leftarrow$ Select top $t_i$ percentage by $F_k^i$;
16    $P^i \leftarrow$ Eq. (4);
17   $F \leftarrow$ Head token fine-tuning using $P^T$;
18   Return $F$;

---

Table 1: Datasets overview.

| Dataset | Classification Type | # Docs | # Classes |
|---------|--------------------|--------|-----------|
| AGNews | News Topic | 120,000 | 4 |
| 20News | News Topic | 17,871 | 5 |
| NYT-Topics | News Topic | 31,997 | 9 |
| NYT-Fine | News Topic | 13,081 | 26 |
| Yelp | Business Review Sentiment | 38,000 | 2 |
| IMDB | Movie Review Sentiment | 50,000 | 2 |
| Amazon | Product Review Sentiment | 3,600,000 | 2 |

$$\mathcal{P}^i = \bigcap_{k=0}^r \mathcal{P}_k^i. \quad (4)$$

The intersection operation can be interpreted as follows: a document and its assigned class belong to $\mathcal{P}^i$ only when it is consistently predicted as the same class and its confidence is ranked top $t_i\%$ by **all** the classifiers $F_k^i$. Therefore, we can ensure to include only those most confident ones into the pseudo label pool to alleviate the error accumulation problem. The less confident predictions of the current iteration will only be left out for the current iteration, but will be re-examined in later iterations and added to pseudo labels if it is qualified.

Finally, we will repeat this iterative process by $T$ full iterations to get the last pseudo label pool $\mathcal{P}^T$. It will then be used for head token fine-tuning of the classifier at iteration $T + 1$ which will be the final classifier of PIEClass. Algorithm 1 summarizes the entire framework.

## 4 Experiments

### 4.1 Experiment Setup

---

[2]See Appx C.4 for some empirical results.

Table 2: Performance of all compared methods measured by Micro-F1/Macro-F1, with the best score **boldfaced** and the second best score underlined. [†] We re-run ClassKG with its official implementation using only the label names for a fair comparison. Other baseline results come from (Meng et al., 2020) and (Wang et al., 2021) with missing values marked as -. [‡] The results are influenced by RoBERTa's tokenizer.

| Methods | AGNews | 20News | NYT-Topics | NYT-Fine | Yelp | IMDB | Amazon |
|---|---|---|---|---|---|---|---|
| **WeSTClass** | 0.823/0.821 | 0.713/0.699 | 0.683/0.570 | 0.739/0.651 | 0.816/0.816 | 0.774/- | 0.753/- |
| **ConWea** | 0.746/0.742 | 0.757/0.733 | 0.817/0.715 | 0.762/0.698 | 0.714/0.712 | -/- | -/- |
| **LOTClass** | 0.869/0.868 | 0.738/0.725 | 0.671/0.436 | 0.150/0.202 | 0.878/0.877 | 0.865/- | 0.916/- |
| **XClass** | 0.857/0.857 | 0.786/0.778 | 0.790/0.686 | 0.857/0.674 | 0.900/0.900 | -/- | -/- |
| **ClassKG**[†] | 0.881/0.881 | 0.811/**0.820** | 0.721/0.658 | 0.889/0.705 | 0.918/0.918 | 0.888/0.888 | 0.926/- |
| **RoBERTa (0-shot)** | 0.581/0.529 | 0.507/0.445[‡] | 0.544/0.382 | -/-[‡] | 0.812/0.808 | 0.784/0.780 | 0.788/0.783 |
| **ELECTRA (0-shot)** | 0.810/0.806 | 0.558/0.529 | 0.739/0.613 | 0.765/0.619 | 0.820/0.820 | 0.803/0.802 | 0.802/0.801 |
| **PIEClass** | | | | | | | |
|   **ELECTRA+BERT** | 0.884/0.884 | 0.789/0.791 | 0.807/0.710 | 0.898/0.732 | 0.919/0.919 | 0.905/0.905 | 0.858/0.858 |
|   **RoBERTa+RoBERTa** | **0.895/0.895** | 0.755/0.760[‡] | 0.760/0.694 | -/-[‡] | 0.920/0.920 | 0.906/0.906 | 0.912/0.912 |
|   **ELECTRA+ELECTRA** | 0.884/0.884 | **0.816**/0.817 | **0.832/0.763** | **0.910/0.776** | **0.957/0.957** | **0.931/0.931** | **0.937/0.937** |
| **Fully-Supervised** | 0.940/0.940 | 0.965/0.964 | 0.943/0.899 | 0.980/0.966 | 0.957/0.957 | 0.945/- | 0.972/- |

**Datasets** We use 7 publicly available benchmark datasets for the weakly-supervised text classification task. Four for news topic classification: **AGNews** (Zhang et al., 2015), **20News** (Lang, 1995), and **NYT-Topics** (imbalanced) and **NYT-Fine** (imbalanced and fine-grained) (Sandhaus, 2008); three for sentiment classification: **Yelp** (Zhang et al., 2015), **IMDB** (Maas et al., 2011), and **Amazon** (McAuley and Leskovec, 2013). Table 1 shows the data statistics, and Table 4 in the Appendix shows the label names and prompt used for each dataset. We follow previous works to use Micro-F1/Macro-F1 as the evaluation metrics.

**Compared Methods** We compare the following methods on the weakly-supervised text classification task: seed-driven methods **WeSTClass** (Meng et al., 2018) and **ConWea** (Mekala and Shang, 2020), which take at least three keywords for each class as input; **LOTClass** (Meng et al., 2020), **XClass** (Wang et al., 2021), and **ClassKG** (Zhang et al., 2021) that only take label names as supervision; two zero-shot prompting baselines **RoBERTa (0-shot)** and **ELECTRA (0-shot)**; and a **Fully-Supervised** BERT baseline as a reference. See more details of compared methods in Appx B.2. We include three versions of PIEClass with different combinations of backbone PLMs:

- **ELECTRA+BERT** uses ELECTRA for prompting and BERT for head token fine-tuning for a fair comparison with baselines using BERT for final classifier training.
- **RoBERTa+RoBERTa** uses RoBERTa as backbone models for the entire framework to compare with baselines using only MLM-based PLM.

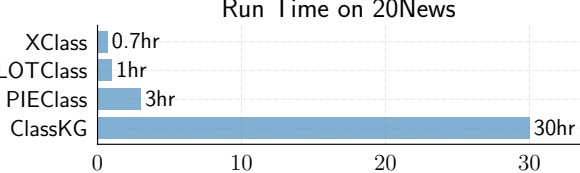

Figure 3: Run time (in hours) on 20News. ClassKG takes much longer time than other methods.

- **ELECTRA+ELECTRA** uses ELECTRA as backbone models for the entire framework.

The implementation details of PIEClass are in Appx B.3.

### 4.2 Experimental Results

Table 2 shows the evaluation results of all methods. PIEClass overall achieves better performance than the compared baselines. It even achieves similar results to the fully-supervised baseline on Yelp and IMDB. We can observe that: (1) ELECTRA+BERT model already outperforms most of the baselines that also use BERT to fine-tune their final classifiers, which shows the effectiveness of our proposed method. (2) ClassKG as an iterative method is the strongest keyword-driven baseline and even achieves better results on 20News than PIEClass. However, it takes a drastically longer time to run. Figure 3 shows the run time on 20News. ClassKG takes more than 30 hours while PIEClass takes only 3 hours to achieve similar results. (3) ELECTRA (0-shot) already achieves comparable results to some simple baselines, confirming our idea that using contextualized text understanding can lead to high-quality pseudo labels. Although RoBERTa (0-shot) does not perform well on AGNews, after the iterative classifier training process, the full

Table 3: Performance of PIEClass (ELECTRA+ELECTRA) and its ablations measured by Micro-F1/Macro-F1.

| Methods | AGNews | 20News | NYT-Topics | NYT-Fine | Yelp | IMDB | Amazon |
|---|---|---|---|---|---|---|---|
| **Two-Stage** | 0.847/0.847 | 0.739/0.733 | 0.776/0.664 | 0.838/0.678 | 0.913/0.913 | 0.870/0.870 | 0.836/0.835 |
| **Single-View ST** | 0.871/0.871 | 0.736/0.737 | 0.757/0.668 | 0.853/0.695 | 0.912/0.912 | 0.846/0.846 | 0.892/0.892 |
| **Co-Training** | 0.877/0.877 | 0.795/0.791 | 0.818/0.715 | 0.877/0.744 | 0.948/0.948 | 0.925/0.925 | 0.930/0.930 |
| **PIEClass** | **0.884/0.884** | **0.816/0.817** | **0.832/0.763** | **0.910/0.776** | **0.957/0.957** | **0.931/0.931** | **0.937/0.937** |

model achieves the best performance, demonstrating the effectiveness of the iterative process of PIEClass. (4) ELECTRA overall performs better than RoBERTa, especially on the sentiment classification task. Also, RoBERTa's performance can be affected by its tokenizer: in 20News, the label name "religion" is separated into two tokens, so we have to use a sub-optimal label name; half of NYT-Fine's label names are tokenized into multiple pieces, so we do not report the performance.

To explain why PIEClass can achieve similar performance to the fully-supervised method, we find that there are some errors in the ground truth labels which could affect the performance of fully-supervised model if used as training data. For example, the following review in Yelp is labeled as negative but predicted as positive by PIEClass: *"My husband had an omelette that was good. I had a BLT, a little on the small side for $10, but bacon was great. Our server was awesome!"*. Because PIEClass only includes the most confident predictions as pseudo labels, it can ensure the quality of its training samples to make the correct prediction.

### 4.3 Ablation Study

To study the effects of each proposed component of PIEClass, we further compare our full model with its three ablations:

- **Two-Stage** is a two-stage version of PIEClass which directly trains the final text classifier using the pseudo labels from zero-shot prompting.
- **Single-View ST (Self-Training)** does not utilize prompt-based fine-tuning as a second view during the iterative process. It thus follows a standard self-training method by using the confident predictions of the head token classifier as the updated pseudo labels for the next iteration.
- **Co-Training** uses the two views of data (i.e., two PLM fine-tuning strategies) to update the pseudo labels in turn with their confident predictions, while in PIEClass the two views are used to regularize each other.

All the compared methods are based on the ELECTRA+ELECTRA version of PIEClass with the same hyperparameters as described in Appx B.3.

Table 3 shows the performance of PIEClass and its ablations on seven datasets. We can observe that: (1) our full model PIEClass consistently outperforms all of its ablations, showing the effectiveness of each ablated component. (2) By removing the iterative pseudo label expansion process, the Two-Stage model performs worse than PIEClass, meaning that the erroneous pseudo labels in the first stage will affect the final classifier training if not corrected. However, the Two-Stage version already achieves comparable results to strong keyword-driven baselines, which shows the power of zero-shot PLM prompting on the text classification task. (3) The Single-View ST model performs similarly to the Two-Stage model and sometimes even worse. This proves that, with the noisy pseudo labels, the standard self-training strategy can cause the error accumulation problem and harm the classifier training. (4) The Co-Training model performs much better than the previous two ablations, meaning that utilizing two PLM fine-tuning methods as two views of data can improve the pseudo label quality. However, it still performs worse than PIEClass, showing that using two views to regularize each other can further improve the noise robustness.

To show the effectiveness of our pseudo label generation and selection, we also compare PIEClass with vanilla few-shot classifiers on AGNews and IMDB. While PIEClass does not need any label, we find that around 500 to 1,000 labels are needed for the few-shot classifiers to achieve similar performance as PIEClass. More details are described in Appx C.1.

### 4.4 Study of the Iterative Process

To study the iterative process of PIEClass, we show the performance of PIEClass and its Single-View Self-Training ablation when varying the number of full iterations from 1 to 5 in Figure 4. From the figure, we can see that, although Single-View Self-Training may perform better than PIEClass when the quantity of pseudo labels is small at the beginning, after five iterations, PIEClass consis-

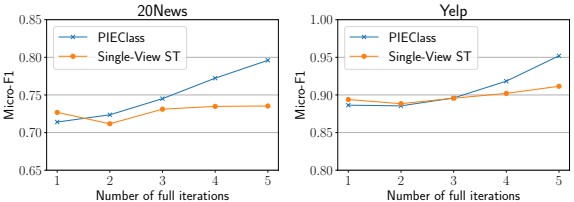

Figure 4: Performance of PIEClass and Single-View ST across varying numbers of full iterations.

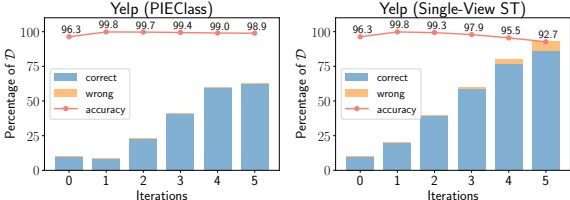

Figure 5: Quantities and qualities of the pseudo labels at each iteration of PIEClass and Single-View ST on Yelp. Each bar shows the portion of correct and wrong pseudo labels to the entire corpus $\mathcal{D}$, and the line shows the quality (accuracy) of pseudo labels.

tently outperforms it. The reason is that the quality of pseudo labels becomes more crucial when the number of pseudo labels increases. Therefore, the performance of Single-View Self-Training does not increase much during the iterative process due to its error accumulation problem, while the performance of PIEClass is increasing much faster. For efficiency, we set the number of iterations to 5 or 3 except for 20News, but running more iterations may further improve the results.

Figure 5 shows the quantity and quality of the pseudo labels at each iteration by PIEClass and Single-View ST on Yelp. The bars represent the percentage of correct and wrong pseudo labels to the entire corpus $\mathcal{D}$, and the lines are their quality measured by accuracy (the number of correct pseudo labels over the total number of pseudo labels). We can observe that, Single-View ST progressively increases the number of pseudo labels but the quality of pseudo labels drops quickly, while PIEClass can keep the quality of pseudo labels during the expansion process. The number of pseudo labels does not increase much in the last two iterations, because PIEClass does not blindly expand the pseudo labels with potential errors. By utilizing two PLM fine-tuning methods and model ensemble, PIEClass only includes the most confident pseudo labels to ensure the quality, which contributes to its superior performance. More results are in Appx C.5.

## 5   Related Work

**Weakly-Supervised Text Classification** Weakly-supervised text classification aims to train a classifier with very limited supervision. Earlier studies utilize distant supervision from knowledge bases such as Wikipedia to interpret the document-label semantic relevance (Gabrilovich and Markovitch, 2007; Chang et al., 2008; Song and Roth, 2014). Some other supervision signals such as keywords (Agichtein and Gravano, 2000; Tao et al., 2018; Meng et al., 2018, 2020; Wang et al., 2021; Zhang et al., 2021) and heuristic rules (Ratner et al., 2016; Badene et al., 2019; Shu et al., 2020) are also explored to reduce the efforts of acquiring any labels or domain-specific data. Recently, the extremely weakly-supervised settings, where only the label name of each class is utilized as supervision, are studied and achieve inspiring results (Meng et al., 2020; Mekala and Shang, 2020; Wang et al., 2021; Zhang et al., 2021). LOTClass (Meng et al., 2020) fine-tunes an MLM-based PLM for category prediction and generalizes the model with self-training. ConWea (Mekala and Shang, 2020) leverages seed words and contextualized embeddings to disambiguate the keywords for each class. XClass (Wang et al., 2021) utilizes keywords to obtain static representations of classes and documents and generates pseudo labels by clustering. ClassKG (Zhang et al., 2021) learns the correlation between keywords by training a GNN over a keyword co-occurrence graph. However, these methods only depend on static keyword features, leading to noisy pseudo-labeled documents for classifier training. LOPS (Mekala et al., 2022) studies the order of pseudo label selection with learning-based confidence scores. A concurrent work MEGClass (Kargupta et al., 2023) studies how different text granularities can mutually enhance each other for document-level classification.

Data programming is another line of work on weak supervision. These methods either require domain knowledge to provide heuristic rules (Chatterjee et al., 2018) or some labeled samples to induce labeling functions (Varma and Ré, 2018; Pryzant et al., 2022), or both (Maheshwari et al., 2021, 2022; Awasthi et al., 2020). In this paper, we focus on training text classifiers with extremely weak supervision, i.e., using only the label names as supervision, so we do not compare with data programming methods that require additional knowledge

like textual patterns and keyword lists (Ren et al., 2020).

**Prompt-Based Learning** PLMs (Devlin et al., 2019; Radford et al., 2019; Liu et al., 2019) have shown superior performance on various downstream tasks through fine-tuning with task-specific data. Some papers show that PLMs can learn generic knowledge during the pre-training stage and design cloze-style prompts to directly probe its knowledge without fine-tuning (Petroni et al., 2019; Davison et al., 2019; Zhang et al., 2020). Later, task-specific prompts are used to guide PLM fine-tuning and perform well in a low-resource setting for several tasks, such as text classification (Han et al., 2021; Hu et al., 2022), relation extraction (Chen et al., 2022), and entity typing (Ding et al., 2021; Huang et al., 2022). Recent works use prompts for keyword or rule discovery (Zeng et al., 2022; Zhang et al., 2022a) or directly prompt large PLMs for weak supervision (Smith et al., 2022). To mitigate the human efforts in prompt engineering, researchers also study automatic methods including prompt search (Shin et al., 2020; Gao et al., 2021) and prompt generation (Guo et al., 2022; Deng et al., 2022). Soft prompts are also proposed by tuning some randomly initialized vectors together with the input (Zhong et al., 2021; Li and Liang, 2021; Lester et al., 2021). Lang et al. (2022) also shows that the co-training method can benefit prompt-based learning in a few-shot setting. Besides standard prompting methods for MLM-based PLMs, prompting methods for discriminative PLMs are also studied on few-shot tasks (Xia et al., 2022; Yao et al., 2022; Li et al., 2022).

## 6 Conclusion and Future Work

In this paper, we study the task of weakly-supervised text classification that trains a classifier using the label names of target classes as the only supervision. To overcome the limitations of existing keyword-driven methods, we propose PIEClass which consists of two modules: (1) an initial pseudo label acquisition module using zero-shot PLM prompting that assigns pseudo labels based on contextualized text understanding, and (2) a noise-robust iterative ensemble training module that utilizes two PLM fine-tuning methods with model ensemble to expand pseudo labels while ensuring the quality. Extensive experiments show that PIEClass can achieve overall better performance than strong baselines, especially on the sentiment classification task where PIEClass achieves similar performance to a fully-supervised baseline.

There are three future directions that can be explored. First, we can extend our method to other forms of text data (e.g., social media) and other abstract classes (e.g., stance detection, morality classification) that require deeper text understanding and keyword-driven methods will likely fail. Second, PIEClass can be integrated with keyword-based methods as two types of training signals to further improve the performance of weakly-supervised text classification. Third, the idea of PIEClass is also generalizable to other text mining tasks with limited supervision, such as named entity recognition and relation extraction.

## Limitations

In this paper, we propose PIEClass, a general method for weakly-supervised text classification. We introduce an iterative ensemble framework by combining two standard PLM fine-tuning methods for noise robustness. Despite its effectiveness shown in the experiments, there is still room for improvement. For example, our learning framework can be integrated with other PLM fine-tuning methods and noise-robust training objectives (Zhang and Sabuncu, 2018; Meng et al., 2021). Besides, our method uses PLM prompting to acquire pseudo labeled documents. As we only use several popular corpora, verbalizers, and prompts for this task, it may require additional efforts to find verbalizers/prompts if working on other domains. Finally, our iterative pseudo label expansion framework requires access to a number of unlabeled documents, so it may perform worse if the corpus is too small.

## Acknowledgments

Research was supported in part by US DARPA KAIROS Program No. FA8750-19-2-1004 and INCAS Program No. HR001121C0165, National Science Foundation IIS-19-56151, IIS-17-41317, and IIS 17-04532, and the Molecule Maker Lab Institute: An AI Research Institutes program supported by NSF under Award No. 2019897, and the Institute for Geospatial Understanding through an Integrative Discovery Environment (I-GUIDE) by NSF under Award No. 2118329. Any opinions, findings, and conclusions or recommendations expressed herein are those of the authors and do not necessarily represent the views, either expressed or implied, of DARPA or the U.S. Government.

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

## A  Preliminaries on PLM Fine-Tuning

Recently, Transformer-based large language models achieve remarkable performance on downstream tasks by first pre-training on large corpora to capture generic knowledge and then fine-tuning with task-specific data. There are generally two fine-tuning strategies for the sequence classification tasks: head token fine-tuning and prompt-based fine-tuning. See Figure 1 for some examples.

**Head Token Fine-Tuning.** PLMs like BERT and RoBERTa add an additional [CLS] token at the beginning of the input sequence and it can be fine-tuned for sequence classification tasks to capture the information of the entire sequence. To fine-tune for a downstream task, the contextualized representation of the [CLS] token $\mathbf{h}^{\text{CLS}}$ of a document $d$ is fed into a linear classification head $g$ to get

$$p(c|d) = \text{Softmax}(g(\mathbf{h}^{\text{CLS}})). \quad (5)$$

Then, given the training samples $\{(d_i, c_i)\}$, the PLM model and the randomly initialized classification head (normally a single linear layer) are optimized with the cross-entropy loss:

$$\mathcal{L}^{\text{head}} = -\sum_i \log p(c_i|d_i). \quad (6)$$

Because the PLM is not pre-trained for any specific downstream task, the [CLS] token embedding does not contain the necessary information if not fine-tuned. Besides, the randomly initialized classification head also needs to be trained. Therefore, normally the head token fine-tuning needs a substantial amount of labeled data for training. Otherwise, the model can easily overfit the training data given a large number of parameters to update. For example, existing weakly-supervised text classification methods use class-indicative keywords to assign pseudo labels to documents which are then used to fine-tune a PLM using its [CLS] token.

**Prompt-Based Fine-Tuning.** To close the gap between PLM's pre-training task and the downstream applications, prompt-based fine-tuning is proposed to convert the input and output of the downstream task to a similar form of the pre-training task. Because of the similarity between pre-training and fine-tuning tasks, prompt-based fine-tuning only needs a small set of samples to achieve competitive performance with head token fine-tuning. For common PLMs pre-trained with masked language modeling (e.g., BERT, RoBERTa), prompt-based fine-tuning uses a template to convert an input sequence into a cloze-style task. Each class also associates with one or more verbalizers, and PLM will predict the likelihood of each verbalizer for the masked position. For example, for a sentiment classification task, a template $\mathcal{T}^{\text{MLM}}$ can transform a document $d$ as:

$$\mathcal{T}^{\text{MLM}}(d) = d \text{ It was [MASK].}$$

Given $\mathcal{T}^{\text{MLM}}(d)$ as input, the pre-trained PLM and its pre-trained MLM head $f$ will generate a probability distribution over its vocabulary, indicating the likelihood of each token appearing in the masked position,

$$p(w|d) = \text{Softmax}(f(\mathbf{h}^{\text{MASK}})). \quad (7)$$

The probability of predicting a class $c$, assuming its label name $l(c)$ as its only verbalizer, is the probability of its verbalizer $p(l(c)|d)$. During fine-tuning, the PLM and its MLM head can be trained with standard cross-entropy loss.

## B Experiment Setup

### B.1 Datasets

Table 4 shows the label names and prompts used for each dataset.

### B.2 Compared Methods

- **WeSTClass** (Meng et al., 2018) trains a CNN classifier with pseudo documents generated based on keyword embeddings and then applies self-training on the unlabeled documents.
- **ConWea** (Mekala and Shang, 2020) utilizes a pre-trained language model to get pseudo labels using contextualized representations of keywords. It then trains a text classifier and uses the results to further expand the keyword sets.
- **LOTClass** (Meng et al., 2020) uses a pre-trained language model to discover class-indicative keywords and then fine-tunes the PLM using self-training with the soft labeling strategy.
- **XClass** (Wang et al., 2021) first expands the class-indicative keyword sets to help estimate class and document representations. Then, a clustering algorithm is used to generate pseudo labels for fine-tuning a text classifier.
- **ClassKG** (Zhang et al., 2021) constructs a keyword graph with co-occurrence relations and self-train a sub-graph annotator, from which pseudo labels are generated for classifier training and the predictions are used to update keywords iteratively.

### B.3 Implementation Details

We use pre-trained ELECTRA-base-discriminator, BERT-base-uncased, and RoBERTa-base as the backbone models for the corresponding versions of PIEClass. The classification head for head token fine-tuning is a single linear layer. The training batch size is 32 for both head token fine-tuning and prompt-based fine-tuning. We train 5 epochs and use AdamW (Loshchilov and Hutter, 2017) as the optimizer for all the fine-tuning tasks. The peak learning rate is $1e-5$ for prompt-based fine-tuning of RoBERTa and $2e-5$ for prompt-based fine-tuning of ELECTRA and all head token fine-tuning, with linear decay. For Yelp and IMDB that have only two classes, to avoid overfitting when the number of pseudo labels is small, we freeze the first 11 layers of the PLM for fine-tuning in the first several iterations and only fine-tune the full model for the final classifier. The model is run on one NVIDIA RTX A6000 GPU. The threshold

for initial pseudo label acquisition is $t^0 = 10\%$. During the iterative process, the coefficient for the increasing size of pseudo labels is $s = 20\%$, except for NYT-Topics and NYT-Fine which are highly imbalanced, for which we set $s = 35\%$ to ensure enough pseudo samples for the rare classes. Notice that this parameter can be decided by just observing the model's intermediate outputs instead of using any labeled data. The threshold of confidence score is $p = 0.95$. We randomly sample $r = 3$ subsets of size $q = 1\%$ of the candidate pseudo labels for prompt-based fine-tuning and model ensemble. We set the number of full iterations $T = 1/s$, which is 5 for AGNews, Yelp, IMDB, and Amazon and 3 for NYT-Topics and NYT-Fine; for 20News that is harder, we run until the number of pseudo labels does not increase, which takes 8 full iterations.

## C Additional Experiments

### C.1 Comparison with Few-Shot Classifiers

We follow similar settings in Meng et al. (2020) and Zhu et al. (2023) to see how many human labeled samples are needed to achieve similar performance as PIEClass. Here, an ELECTRA-base is fine-tuned with a few labeled samples in a fully-supervised way and compared with PIEClass (ELECTRA+ELECTRA). Since PIEClass does not use any annotated validation data under the weakly-supervised setting, for a fair comparison, we do not provide validation sets for the few-shot classifiers as well. Instead, we directly report the results of the last checkpoint in the training process and we do not see obvious model overfitting on the training set. The results in Macro-F1 on AGNews and IMDB are shown in Table 5. We can see that it needs around 150 labels per class on AGNews (totally 600) and 500 labels per class on IMDB (totally 1000) to achieve similar results to PIEClass, which requires a non-trivial amount of labeling efforts. In contrast, PIEClass only needs one label name for each class as supervision and thus drastically reduces the needs of human efforts.

### C.2 Impact of Backbone PLMs

We include the ELECTRA+BERT version to compare with baselines that also use BERT as the backbone model for classifier training, and also the RoBERTa+RoBERTa version which only uses one single MLM-based model without access to ELECTRA. Also, ELECTRA is pre-trained with the same data as BERT, so using it does not give the model

Table 4: Label names and prompts used for each dataset.

| Dataset | Label Names | Prompt |
|---|---|---|
| AGNews | politics, sports, business, technology | [MASK] News: <doc> |
| 20News | computer, sports, science, politics, religion | [MASK] News: <doc> |
| NYT-Topics | business, politics, sports, health, education, estate, arts, science, technology | [MASK] News: <doc> |
| NYT-Fine | music, baseball, business, abortion, military, football, television, economy, dance, soccer, cosmos, surveillance, golf, law, basketball, budget, movies, stocks, gun, energy, environment, hockey, healthcare, immigration, tennis, gay | [MASK] News: <doc> |
| Yelp | good, bad | <doc> It was [MASK]. |
| IMDB | good, bad | <doc> It was [MASK]. |
| Amazon | good, bad | <doc> It was [MASK]. |

Table 5: Macro-F1 of vanilla few-shot classifiers with different numbers of labels per class compared with PIEClass (ELECTRA+ELECTRA).

| # labels per class | 100 | 150 | 250 | 500 | PIEClass |
|---|---|---|---|---|---|
| AGNews | 0.875 | 0.885 | - | - | 0.884 |
| IMDB | 0.900 | 0.915 | 0.925 | 0.933 | 0.931 |

Table 6: Performance of original XClass, XClass with ELECTRA as the final classifier, and two versions of PIEClass measured by Micro/Macro-F1.

| Methods | AGNews | 20News | NYT-Topics | Yelp |
|---|---|---|---|---|
| XClass | 0.857/0.857 | 0.786/0.778 | 0.790/0.686 | 0.900/0.900 |
| XClass-ELECTRA | 0.838/0.837 | 0.792/0.784 | 0.787/0.685 | 0.903/0.903 |
| PIEClass | | | | |
|   ELECTRA+BERT | **0.884/0.884** | 0.789/0.791 | 0.807/0.710 | 0.919/0.919 |
|   ELECTRA+ELECTRA | **0.884/0.884** | **0.816/0.817** | **0.832/0.763** | **0.957/0.957** |

explicit advantages. Both versions achieve strong enough performance on all datasets, especially on sentiment classification, which demonstrates the effectiveness of PIEClass. We did not include a BERT+BERT version because RoBERTa, as a powerful variant of BERT, is used more popularly for prompting with MLM.

We also run XClass by using ELECTRA as the backbone of its final classifier to study the effects. Also note that we only change the backbone of the final classifier of XClass and still use BERT for its pseudo label assignment step, because we empirically find that switching to ELECTRA for its entire framework drastically decreases its performance: we get 0.77 on Yelp and only 0.43 on AGNews. This also shows that XClass is not generalizable to different types of PLMs while PIEClass is applicable to various types of PLMs as shown in our experiments. Table 6 shows the Micro/Macro-F1 scores, with ELECTRA bringing small improvements to XClass's performance on two datasets and PIEClass (ELECTRA+ELECTRA) consistently outperforms XClass and XClass-ELECTRA.

## C.3 Parameter Sensitivity

We study the parameter sensitivity of PIEClass by varying the following parameters on IMDB: the threshold $t^0$ for the initial pseudo labels, the minimum probability threshold $p$ during the iterations, and the number of prompt-based classifiers $r$. Figure 6 shows performance measured by Micro-F1. We find that overall PIEClass is not sensitive to these parameters.

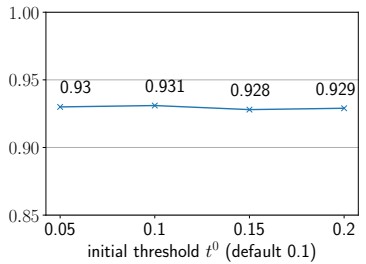 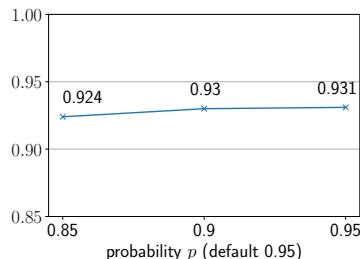 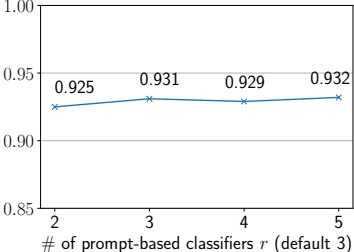

Figure 6: Performance of PIEClass on IMDB when varying different parameters.

Table 7: Macro-F1 on AGNews with a constant threshold $t_i$ =20% for pseudo label selection.

| Iteration | 1 | 2 | 3 | 4 | 5 |
|---|---|---|---|---|---|
| **Macro-F1** | 0.847 | 0.815 | 0.775 | 0.780 | 0.780 |

### C.4 Effects of Using a Constant Threshold $t_i$

We increase the threshold $t_i$ in each iteration to gradually increase the number of selected confident pseudo labels during the iterative process. Because using a constant threshold can make the pseudo samples almost the same in the iterative process and the classifiers overfit more to the limited number of pseudo samples. We tried on AGNews by keeping the threshold constantly equal to 20%. The performance of the classifier in each iteration is shown in Table 7, where we can see the performance first drops and then stays almost constant. In this work, we simply use a linearly increasing threshold. In fact, more advanced curriculum learning strategies can be applied to better fit the distribution of prediction scores, e.g., a self-pacing function (Pei et al., 2022).

### C.5 Additional Results on the Study of Iterative Process

Figure 7 and Figure 8 show more results for studying the iterative process of PIEClass (c.f. Sec 4.4). We can observe similarly that PIEClass can ensure higher quality pseudo labels during the iterative process compared with Single-View ST.

## D Discussions on PLM Prompting

**Handling Multi-Token Label Names** As shown in the experiment results (Table 2), the performance of prompting with MLM-based PLMs such as RoBERTa is affected by the tokenizer, because the MLM classification head cannot naturally handle verbalizers (i.e., label names) with multiple

Table 8: Performance of PIEClass and zero-shot prompting of ELECTRA with different sets of verbalizers, measured by Micro-F1/Macro-F1.

| Verbalizers | Methods | Yelp | IMDB |
|---|---|---|---|
| good/bad | **ELECTRA (0-shot)** | 0.820/0.820 | 0.803/0.802 |
| | **PIEClass** | 0.957/0.957 | 0.931/0.931 |
| great/ terrible | **ELECTRA (0-shot)** | 0.880/0.880 | 0.844/0.844 |
| | **PIEClass** | 0.959/0.959 | 0.933/0.933 |

tokens. For example, the label name "religion" of 20News is tokenized by RoBERTa into two tokens, "rel" and "igion". Therefore, prompting RoBERTa for multi-token label names requires substantially more work by inserting multiple [MASK] tokens into the template and iteratively predicting the masked tokens. On the other hand, prompting ELECTRA can easily handle multi-token label names (Xia et al., 2022), because the label names are directly encoded in the input. Assume that a label name $l(c)$ is tokenized into several pieces $l(c) = \{w_1, \dots, w_{|l(c)|}\}$. We can estimate the probability of its being original by taking the average of the probabilities of each token,

$$p(l(c)|d) = \frac{1}{|l(c)|} \sum_{i=1}^{|l(c)|} p(w_i|d) \qquad (8)$$

**Templates for PLM Prompting** One limitation of PLM prompting is that its performance is related to the quality of the templates and verbalizers. In this work, we directly use the prompts for sentiment classification and news topic classification from previous studies (Gao et al., 2021; Xia et al., 2022) without any further tuning. To mitigate the human efforts on prompt engineering, some automatic methods are proposed to optimize the prompts, including prompt search (Shin et al., 2020; Gao et al., 2021) and prompt generation (Guo et al., 2022; Deng et al., 2022).

The selection of verbalizers can also affect the performance of prompt-based methods for PLMs.

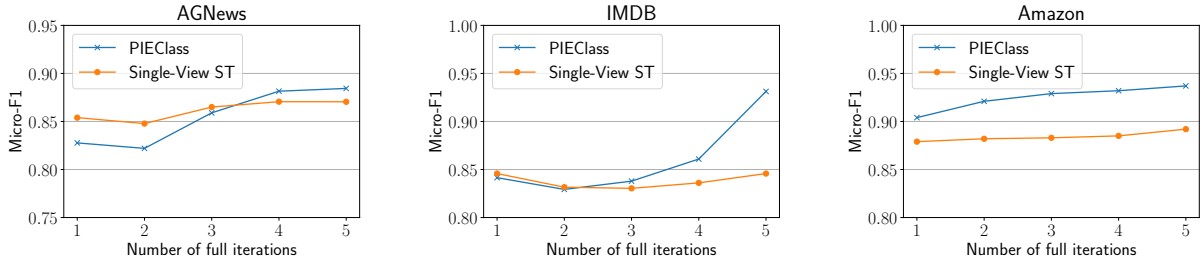

Figure 7: Performance of PIEClass and Single-View ST by varying the number of full iterations.

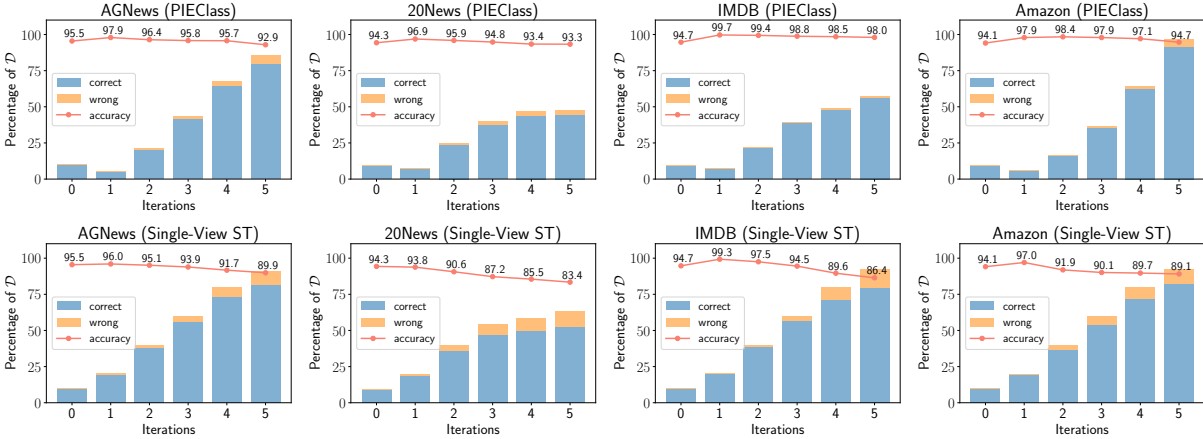

Figure 8: Quantities and qualities of the pseudo labels at each iteration of PIEClass (top) and the Single-View ST ablation (bottom).

In this paper, we directly use the label names from previous works on weakly-supervised text classification as our main results. Here, we also try a different set of verbalizers for sentiment classification, "great" and "terrible", that are used in previous papers studying prompt-based methods (Gao et al., 2021; Xia et al., 2022). Table 8 shows the performance of ELECTRA (0-shot) and PIEClass with the two sets of verbalizers. We can see that, by changing the verbalizers, the zero-shot prompting performance increases by a large amount and even achieves comparable results to the keyword-driven baselines on Yelp. PIEClass also performs better with the new verbalizers. Therefore, optimizing verbalizers could be a promising next step for prompt-based text classification by verbalizer search (Gao et al., 2021; Schick et al., 2020) or learning verbalizer-class correlation (Huang et al., 2022).