# OpenReview forum: "PIEClass: Weakly-Supervised Text Classification with Prompting and Noise-Robust Iterative Ensemble Training"
_EMNLP/2023/Conference — EMNLP 2023 Main_

### Official Review · Reviewer_cc9r · 2023-08-03

**Soundness:** 4

**Excitement:**

3: Ambivalent: It has merits (e.g., it reports state-of-the-art results, the idea is nice), but there are key weaknesses (e.g., it describes incremental work), and it can significantly benefit from another round of revision. However, I won't object to accepting it if my co-reviewers champion it.

**Paper Topic And Main Contributions:**

For weakly supervised text classification, this paper proposes to apply the zero-shot prompting of PLM to obtain pseudo labels. To alleviate the noise problem induced by the pseudo label generation, this paper designs an iterative ensemble training method, which utilizes two PLM fine-tuning strategies and a model ensemble strategy to improve the quality of the pseudo label.  Experimental results show that the proposed method can achieve SOTA on seven datasets, and the performance is close to that of the fully-supervised baseline in two datasets.

**Questions For The Authors:**

A. In the prompting-based fine-tuning phase, the dataset is split into multiple subsets to train multiple classifiers respectively. With little training sample, is it possible to cause overfitting of a single classifier and reduce its robustness?
B. Why is the threshold t_i set to increase linearly?
C. Why does the ELECTRA+ELECTRA method perform so poorly in the YELP dataset in Table 5?
D. It seems that the benefits of the proposed pseudo label generation method are not analyzed in the experiment?


**Reasons To Accept:**

1. The noise problem solved in this paper is important for Weakly-Supervised learning, which is a common dilemma when applying pseudo label generation.
2. This article is well organized and is well illustrated to help the reader understand.
3. The experiments designed in this paper are sufficient, not only to design ablation experiments for exploring different components of the model, but also to explore the change of pseudo label quality during the iteration process.
4. The idea of using two PLM fine-tuning methods for co-training is interesting, and the performance improvements are also obvious.


**Reasons To Reject:**

1.The reasons for some detail design in the method are not well explained.

**Reproducibility:**

4: Could mostly reproduce the results, but there may be some variation because of sample variance or minor variations in their interpretation of the protocol or method.

**Reviewer Confidence:**

4: Quite sure. I tried to check the important points carefully. It's unlikely, though conceivable, that I missed something that should affect my ratings.

---

> ### Author Rebuttal · Authors · 2023-08-28
>
> Thanks for your comments on our paper and your recognition of the novelty and significance of our work! We are also glad to know you find our paper well-organized and easy to understand. We address the questions you raised as follow:
>
> **1. Is it possible to have overfitting of a single classifier**
> Thanks for raising this question. Yes, it is possible that given the small number of pseudo samples, a single classifier may overfit, and empirically we find that even without subsampling but using all pseudo labels for training, the model still can overfit in the earlier iterations given the small number of labels. This actually motivates us to **apply model ensembling to improve robustness against overfitting**. By randomly sampling subsets of pseudo labels and training multiple individual classifiers, it is unlikely that all classifiers overfit to the same set of samples/features. Then, by taking the intersection of the top predictions of all classifiers, we can largely mitigate the impact of potential noise caused by overfitted models. This can also be shown in our experiment analysis that our method can keep the pseudo label quality during the iterative process and outperform the Co-training ablation which also utilizes two PLM fine-tuning methods but does not apply model ensemble.
>
> **2. Why the threshold $t_i$ increases linearly**
> We increase the threshold in each iteration to gradually increase the number of selected confident pseudo labels during the iterative process. This is because if we keep the threshold constant, the pseudo samples in the last iteration will be predicted confidently again, making the pseudo samples almost the same in the iterative process and the classifiers overfit more to the limited number of pseudo samples. We tried on AGNews by keeping the threshold constantly equal to 20%. The performance of the classifier in each iteration is shown in the following table, where we can see the performance first drops and then stays almost constant.
> | **Iteration #** | **1**| **2** | **3** | **4** | **5** |
> | :-- |:--|:--|:--|:--|:--|
> |**Macro-F1 w/ constant $\mathbf{t}$**| 0.847 | 0.815 | 0.775 | 0.780 |0.780|
>
> In this work, we simply use a linearly increasing threshold. In fact, more advanced curriculum learning strategies can be applied to better fit the distribution of prediction scores, e.g., a self-pacing function [1]. Here, we try to keep it simple as it is not the major claim of our work.
>
> **3. Low performance score in Table 5**
> It is a typo and thanks for catching it. The actual performance of ELECTRA+ELECTRA on Yelp should be 0.957, which can also be seen in the main results Table 2. We will correct it in the revision.
>
> **4. Benefits of pseudo label generation**
> Overall, we use PLM prompting for pseudo label generation which is contextualized given the prompts and can go beyond static keyword-based features. The benefits of the proposed pseudo label generation can be seen in the following two parts in the experiment results. **First**, in Table 2, we show the performance of zero-shot PLM prompting as two baselines and can observe that prompting ELECTRA can already achieve strong performance on some datasets. For example, on NYT-Fine, it achieves 0.765 micro-F1 which is better than the simple weakly-supervised baselines. **Second**, we report the performance of the Two-Stage ablation in Table 3, which is directly fine-tuning the final classifier on the initial pseudo labels. We can see it achieves comparable results to the strong baselines on all datasets, which shows the effectiveness of the proposed pseudo label generation step.
>
> We would like to thank you again for your comments! We appreciate your recognition of the importance of our paper. We hope our rebuttal addresses your questions and we are happy to answer any further questions you may have.
>
> [1] Pei et al., Graph Alignment with Noisy Supervision, WWW 2022.

---

### Official Review · Reviewer_dUCf · 2023-08-05

**Soundness:** 4

**Excitement:**

3: Ambivalent: It has merits (e.g., it reports state-of-the-art results, the idea is nice), but there are key weaknesses (e.g., it describes incremental work), and it can significantly benefit from another round of revision. However, I won't object to accepting it if my co-reviewers champion it.

**Missing References:**

please see the references mentioned reasons to reject.

**Paper Topic And Main Contributions:**

This work proposes PIEClass, a zero-shot text classification approach that performs well on seven benchmark datasets. This paper aims to train a text classifier that categorizes a given text into one of the predefined classes without using any text-label pairs for training (hence zero-shot). PIEClass iteratively trains the classifier. In each iteration, two models jointly determine a set of examples, together with their pseudo-labels (similar to self-training). These selected examples will be used for training the models in the next iteration. The key contribution lies in combining powerful methods to push the limit of zero-shot text classification.

**Questions For The Authors:**

1. Question A: this question is related to the first two reasons to reject. Have the authors compared performance with vanilla fine-tuning on a few labeled samples? Given that iterative training increases the training time (Figure 3), what if one chooses vanilla fine-tuning for fast training and devotes the saved time to manually annotating a small set of samples? What would be the performance gap to zero-shot learning methods?
2. Question B:  at each iteration, is the F^{i}_{0} (L325) always freshly initialized? Or do the authors continue training the model from the previous iteration?

**Reasons To Accept:**

1. The proposed method works well on various datasets, achieving strong performance without training on annotated data. This should establish a strong baseline for future work in zero-shot text classification.
2. The paper is well-written and easy to follow, and the techniques used are sound.
3. Overall, the experiments and results are solid.

**Reasons To Reject:**

1. The practicability of this method is still somewhat questioned. There is a set of hyperparameters that need to be determined to apply the proposed method. It is unclear how to determine these values in practice. While appendix C2 discusses the parameter sensitivity, it is only done on one dataset (IMDB). It needs to be clarified about the sensitivity of other datasets. In addition, appendix B3 (L971 - L981) mentions that different values are needed for different datasets. These values can only be determined by accessing certain information about the datasets (e.g., class imbalance). On the other hand, the effort of getting this information (e.g., by manually checking some samples from the dataset) may instead be devoted to getting some labeled samples for training.
The concern of practicability can be addressed in either of the two ways. a) use a fixed set of hyperparameters across all datasets and report the results again. This would show that the model is not sensitive to the hyperparameters across all settings. b) use a small set of labeled data for validation purposes so that the optimal hyperparameters can be decided [1][2][3]. But then one should also compare with a simple baseline that trains on this set of labeled data [3].
2. An additional aspect regarding practicability: the paper mentions that “Earlier studies train text classifiers in a fully -supervised manner that requires a substantial amount of training data” (L38-39). Is that true for PLMs like RoBERTa? The paper only contrasts their performance with models that trained on all samples (the fully supervised upper bound) but not with models that trained on a small set of samples (i.e., still supervised learning, but in a few-shot learning setting).
3. In Table 2 & 3: the standard deviation should also be reported, given that the performance gaps among top-performing systems are small. There would be more space if the authors reported performance from a range of 1-100 (88.4 vs. 0.884). If there is still a lack of space, the authors can consider reporting only Marco-F1 in the main paper and moving Micro-F1 to the appendix. Also, the number of runs for each experiment should be reported.

[1] Oliver et al., Realistic Evaluation of Deep Semi-Supervised Learning Algorithms
[2] Perez et al., True Few-Shot Learning with Language Models
[3] Zhu et al., Weaker Than You Think: A Critical Look at Weakly Supervised Learning

**Reproducibility:**

4: Could mostly reproduce the results, but there may be some variation because of sample variance or minor variations in their interpretation of the protocol or method.

**Reviewer Confidence:**

4: Quite sure. I tried to check the important points carefully. It's unlikely, though conceivable, that I missed something that should affect my ratings.

---

> ### Author Rebuttal · Authors · 2023-08-28
>
> Thanks for your comments on our paper! We are happy to know that you find our paper easy to follow and appreciate your recognition of the significance of our paper for weakly supervised and zero-shot text classification.
>
> We would like to address your concerns on our paper as follows
>
> **1. Hyperparameter setting**
> Thanks for your constructive comments. We agree with you that using the same set of hyperparameters across all datasets makes our evaluation better align with practical scenarios. However, because the compared baselines also use different hyperparameters for different datasets, e.g., the number of epochs in LOTClass ([github](https://github.com/yumeng5/LOTClass)), the number of keywords per class and batch sizes in ClassKG ([github](https://github.com/zhanglu-cst/ClassKG)), we believe it is unavoidable for a fair comparison. The only parameter that is different in our experiments is the coefficient $s$ for the increasing pseudo label size, which is 35% for two NYT and 20% for others. This number is decided because of the unbalancing nature of the NYT corpus, and having enough pseudo samples for the rare classes can help the training process. In fact, unlike hyperparameters such as batch size or learning rate that do require labeled samples for tuning, **the hyperparameter $s$ can be tuned just by observing the model’s intermediate outputs instead of using labeled data**. For example, for the NYT-Fine dataset which is unbalanced across classes, if we set $s=20$%, the rare classes will have less than 3 pseudo labeled samples (only 1 for the most infrequent class). Therefore, we increase $s$ to 35% which gives at least 10 pseudo samples per class.
>
> For the parameter sensitivity analysis in Appendix C.2, we observe similar trends on other datasets and thus select one dataset IMDB to illustrate it. We will clarify this in the revision.
>
> We also follow your suggestion (a) and run PIEClass (ELECTRA+ELECTRA) on two NYT datasets with the same parameters as other datasets (i.e., $s=20$%). We compare its results with other extremely weakly-supervised methods in the following table (thus excluding WeSTClass and ConWea). Although the performance is lower than the original setting, PIEClass still achieves strong performance compared with other extremely weakly-supervised baselines.
> | **Methods** | **NYT-Topics** | **NYT-Fine** |
> |:-------------|:----------------|:---------------|
> |**LOTClass**|   0.671/0.436  |  0.150/0.202  |
> | **XClass**   |  0.790/0.686   |  0.857/0.674  |
> | **ClassKG**|   0.721/0.658  |  0.889/0.705  |
> | **PIEClass (s=20%)**|  0.758/0.719   |  0.898/0.689  |
>
> **2. Comparison with a few-shot learner**
> We agree with you that comparing a few-shot baseline with vanilla PLM fine-tuning is worth investigating. We thus follow similar settings in [1] and [2] to see how many human labeled samples are needed to achieve similar performance as PIEClass. Here, a PLM is fine-tuned with a few labeled samples in a fully-supervised way, and we use ELECTRA-base as the backbone model for a fair comparison with PIEClass (ELECTRA+ELECTRA). Since PIEClass does not use any annotated validation data under the weakly supervised setting, for a fair comparison, we do not provide validation sets for the few-shot classifiers as well. Instead, we directly report the results of the last checkpoint in the training process and we do not see obvious model overfitting on the training set. We perform 3 individual runs and take the average score to account for the effect of randomness in label sampling. The results in Macro-F1 ($\pm$ standard deviation) on AGNews and IMDB are shown in the following table. We can see that it needs around 150 labels per class on AGNews (totally 600) and 500 labels per class on IMDB (totally 1000) to achieve similar results of PIEClass, which requires a non-trivial amount of labeling efforts. In contrast, PIEClass only needs one label name for each class as supervision and thus drastically reduces the needs of human efforts.
> | **# labels per class** | **100** | **150** | **250** | **500** | **PIEClass**|
> |:--|:--|:--|:--|:--|:--|
> |**AGNews**|87.5 ($\pm$ 0.45)|88.5 ($\pm$ 0.26)|-|-|88.4 ($\pm$ 0.21)|
> |**IMDB**|90.0 ($\pm$ 1.23)|91.5 ($\pm$ 0.91)|92.5 ($\pm$ 0.43)|93.3 ($\pm$ 0.25)|93.1 ($\pm$ 0.09)|
>
> **3. Standard deviation**
> While many previous studies also do not report standard deviation, we agree that it is an important indicator on measuring the stability. Thanks for your suggestions on the formatting and we will take that into consideration in the revision. Empirically, we find that running PIEClass with different seeds only has very small influence on the final performance. With three individual runs of PIEClass (ELECTRA+ELECTRA) on IMDB, we get an averaged Macro-F1 score of 0.931 with standard deviation 0.00094.
>
> **4. Model initialization in each iteration**
> The classifiers are freshly initialized with the pre-trained weights and not continuously trained from the last iteration. We will clarify it in the revision.
>
> We would like to thank you again for your constructive comments! Hope we have addressed your concerns and we are happy to answer any further questions you may have.
>
> [1] Zhu et al., Weaker Than You Think: A Critical Look at Weakly Supervised Learning, ACL 2023.
> [2] Meng et al., Text classification using label names only: A language model self-training approach, EMNLP 2020.

---

### Official Review · Reviewer_oKTC · 2023-08-07

**Soundness:** 3

**Excitement:**

3: Ambivalent: It has merits (e.g., it reports state-of-the-art results, the idea is nice), but there are key weaknesses (e.g., it describes incremental work), and it can significantly benefit from another round of revision. However, I won't object to accepting it if my co-reviewers champion it.

**Missing References:**

The data programming also corresponds to the weak-supervision literature. None of the relevant papers from the data-programming domain seems to be discussed and/or compared in the paper. Refer few papers below:

a) Robust Data Programming with Precision-guided Labeling Functions. AAAI 2020: 3397-3404

b) Learning to Robustly Aggregate Labeling Functions for Semi-supervised Data Programming. ACL (Findings) 2022: 1188-1202

c) Semi-Supervised Data Programming with Subset Selection. ACL/IJCNLP (Findings) 2021: 4640-4651

d) Learning from Rules Generalizing Labeled Exemplars. ICLR 2020

The similar iterative process is adopted by Snuba for selecting the most relevant rules for the text classification.
e) Snuba: Automating Weak Supervision to Label Training Data, VLDB 2018

f) Automatic Rule Induction for Efficient Semi-Supervised Learning, EMNLP 2022

**Paper Topic And Main Contributions:**

The paper proposes a weak-supervision technique wherein keywords (or deep features)  are leveraged for classification. The paper proposes to rely on output labels instead of the input sequence to guide the classification pipeline. It acquires labels via pre-trained language (PLM) model prompting. It performs an iterative process to train using pseudo labels by PLM and selects most confident predictions and trains a text classifier on those instances. To avoid propagation of incorrect labeling by PLMs, it use an ensemble of two PLM based classifier  as the final prediction.




**Questions For The Authors:**

1. It seems there is a significant overlap with the data programming literature. However, I cannot find its mention in the paper except Line 555.

2. A complete 'Algorithm' describing the whole process would be helpful to understand. Line 324-336 describes but it can be more useful in the form of an algorithm.

3. For the uninitiated, head-token fine-tuning should be explained properly in the main paper itself. Figure 1 caption attempts to explain but IMO it should be described more formally.

4. Line 325 : "text classifier, $F_0^i : D → C$,using the current pseudo labels $P^{i−1}$ in a fully- supervised way" . Should it be $P^i$ ?
In the same paragraph, what does $0$ in  $F_0^i$ refers to ?

5. In Line 340, for the sake of clarity, it should be mentioned that  $q \in [0,1]$. In line 339, is random sampling done with a replacement ?

6. Line 339: "Because the noisy labels are unlikely to be sampled repeatedly into different subsets". Is it because you are selecting only top-k confident predictions from each classifier ?

7. The paper should contain atleast some technical details about ELECTRA in the main paper itself since its a key component for reducing noise by PLMs.

8. What happens to the less-confident predictions in further iterations. Are those instances dropped from the set ?

**Reasons To Accept:**

1. The paper presents an interesting approach to avoid noise from PLMs using an ensemble approach.

**Reasons To Reject:**

1. The methodology section is difficult to understand. (See Qs for the Authors).

2. Discussion of related work seems to miss some important weak-supervision literature.

**Reproducibility:**

3: Could reproduce the results with some difficulty. The settings of parameters are underspecified or subjectively determined; the training/evaluation data are not widely available.

**Reviewer Confidence:**

5: Positive that my evaluation is correct. I read the paper very carefully and I am very familiar with related work.

---

> ### Author Rebuttal · Authors · 2023-08-28
>
> Thanks for your constructive comments on our paper! We address your questions as follows and will clarify them in the revision:
>
> **1. Missing discussion to data programming works:**
> Thanks for pointing out this line of work. We did not discuss data programming methods in detail because they either **require domain knowledge** [1] to provide heuristic rules or **some labeled samples** [2, 3] to induce labeling functions (or both [4, 5, 6]). Because we focus on training text classifiers with extremely weak supervision, i.e., **using only the label names as supervision**, we did not compare our method with the data programming methods that require additional knowledge as supervision (e.g., keyword lists for AGNews, textual patterns for IMDB, and keywords from multiple views/aspects for Yelp [7]). Although Snuba [2] also introduced an iterative learning process, it is under the semi-supervised setting where a set of labels is available and its iterative feedback is used for generating more heuristics.
>
> We agree with you that this line of work is related to the weak supervision setting discussed in our paper, and **we will add related discussion in the revision**. In fact, our method can also be considered as using PLM prompting as the labeling function which requires minimal amount of knowledge for prompt designing, compared with the heuristic rules and/or labeled samples for data programming that require some level of domain expertise. We propose an iterative ensemble approach that can effectively mitigate the impact of noisy labels for classifier training in the extremely weakly supervised setting without any labeled data.
>
> **2. Add a complete ‘Algorithm’:**
> Due to the space limitation, we did not put an ‘Algorithm’ or pseudocode in the methodology section. We agree with you that having such can help readers better understand the methodology and we will add it in the revision.
>
> **3. Discussion on uninitialized head-token fine-tuning:**
> We have more detailed discussion on the preliminaries of head-token fine-tuning and prompt-based fine-tuning in Appendix A due to the space limitation of the main content. We will move those into the main content when possible in the revision.
>
> **4. Notations in Line 325-326:**
> $P^{i-1}$ is defined as the pseudo label pool constructed in iteration $i-1$, so it is also the pseudo label available at the starting of iteration $i$ and is thus used as the training label for the classifier $F^i_0$. The 0 in the subscription of $F^i_0$ is just to represent the head-token classifier in each iteration and we can also name it as $F^i_{head}$. However, since later we will ensemble its results with $r$ prompt-based classifiers which we name as $F^i_1$ to $F^i_r$, we therefore give it the number zero so that (1) it shows the head-token fine-tuning happens before prompt-based fine-tuning and (2) the consistency of numbering makes it cleaner in the ensemble step Eq 4.
>
> **5. Parameter q and random sampling:**
> Each random sampling is done without replacement. Thanks for pointing this out and we will clarify it in the revision.
>
> **6. Line 339, noise during random sampling:**
> It is because the sampling ratio (i.e., $q$) is kept small and the ratio of noisy labels is small compared with the correct labels as we only take top predictions to ensure quality.
>
> **7. Technical details of ELECTRA:**
> We agree that some technical details of ELECTRA, or in general discriminative PLMs, will better explain our methodology. Due to the limited space, we currently have an overall introduction to ELECTRA (Line 221-231) and citations to the original paper and related papers that use prompting with ELECTRA (see Clark et al., 2020, Xia et al., 2022, Yao et al., 2022, Li et al., 2022, in the submitted paper). We will add more details in the main content in the revision if there is space.
>
> **8. Less-confident predictions:**
> The less confident predictions of the current iteration will only be left out from the pseudo label pool of current iteration, but will be re-examined in later iterations and added to later pseudo labels if it is qualified. Because throughout the iterative process, as more quality pseudo labels are available for classifier training, we expect the classifier can make better predictions during this process and wrongly predicted samples in earlier iterations have a chance to be corrected with more powerful classifiers in later iterations.
>
> We would like to thank you again for your comments on our paper and we hope our clarifications help you better understand our method. We will also clarify the points you raised in the revision. We are happy to answer any remaining questions you may have.
>
> [1] Chatterjee et al., Data Programming Using Continuous and Quality-Guided Labeling Functions, in AAAI 2020.
> [2] Varma and Re, Snuba: Automating Weak Supervision to Label Training Data, in VLDB 2018.
> [3] Pryzant et al., Automatic Rule Induction for Efficient Semi-Supervised Learning, in EMNLP 2022.
> [4] Maheshwari et al., Semi-Supervised Data Programming with Subset Selection, in Findings of ACL 2021.
> [5] Maheshwari et al., Learning to Robustly Aggregate Labeling Functions for Semi-supervised Data Programming, in Findings of ACL 2022.
> [6] Awasthi et al., Learning from Rules Generalizing Labeled Exemplars, in ICLR 2020.
> [7] Ren et al., Denoising Multi-Source Weak Supervision for Neural Text Classification, in Findings of EMNLP 2020.

---

### Meta-Review · Area_Chair_7yfa · 2023-09-19

**Recommendation:** 5

**Metareview:**

Reviewers are cautiously positive in their assessment of the paper (3-4).  The proposed method of quickly and cheaply creating labels can be applied to other tasks outside of those tested in this paper.   The agreed upon weakness of the paper is an unclear methodological description, which can be resolved in revisions.  The authors made an effort to address concerns by reviewers.

---

### Decision · Program_Chairs · 2023-10-07

**Decision:**

Accept-Main

**Comment:**

Reviewers are cautiously positive in their assessment of the paper (3-4).  The proposed method of quickly and cheaply creating labels can be applied to other tasks outside of those tested in this paper.   The agreed upon weakness of the paper is an unclear methodological description, which can be resolved in revisions.  The authors made an effort to address concerns by reviewers.